# Recognition of 3D Images by Fusing Fractional-Order Chebyshev Moments and Deep Neural Networks

**DOI:** 10.3390/s24072352

**Published:** 2024-04-07

**Authors:** Lin Gao, Xuyang Zhang, Mingrui Zhao, Jinyi Zhang

**Affiliations:** 1School of Information Science and Engineering, Shenyang Ligong University, Shenyang 110159, Chinazhangjinyi@sylu.edu.cn (J.Z.); 2School of Computer Science and Engineering, Northeastern University, Shenyang 110169, China; 3School of Mechanical Engineering, Shenyang Ligong University, Shenyang 110159, China; 4Faculty of Engineering, Gifu University, Gifu 501-1193, Japan

**Keywords:** fractional order, deep neural network, Chebyshev moments, image recognition

## Abstract

In order to achieve efficient recognition of 3D images and reduce the complexity of network parameters, we proposed a novel 3D image recognition method combining deep neural networks with fractional-order Chebyshev moments. Firstly, the fractional-order Chebyshev moment (FrCM) unit, consisting of Chebyshev moments and the three-term recurrence relation method, is calculated separately using successive integrals. Next, moment invariants based on fractional order and Chebyshev moments are utilized to achieve invariants for image scaling, rotation, and translation. This design aims to enhance computational efficiency. Finally, the fused network embedding the FrCM unit (FrCMs-DNNs) extracts depth features to analyze the effectiveness from the aspects of parameter quantity, computing resources, and identification capability. Meanwhile, the Princeton Shape Benchmark dataset and medical images dataset are used for experimental validation. Compared with other deep neural networks, FrCMs-DNNs has the highest accuracy in image recognition and classification. We used two evaluation indices, mean square error (MSE) and peak signal-to-noise ratio (PSNR), to measure the reconstruction quality of FrCMs after 3D image reconstruction. The accuracy of the FrCMs-DNNs model in 3D object recognition was assessed through an ablation experiment, considering the four evaluation indices of accuracy, precision, recall rate, and F1-score.

## 1. Introduction

Image recognition is an important field of artificial intelligence research, and different forms of moments are key descriptors for extracting relevant information in 3D images. The study of image moments has aroused strong interest among researchers. Moments are widely applied in image reconstruction [1,2,3], image analysis, image indexing [4,5,6,7], digital image research [8,9,10], spectral image super-resolution mapping [11], hyperspectral target detection [12], radar target recognition [13,14], SAR target recognition [15], sound classification [16], and other fields. Li et al. [17] employed an innovative face recognition method that integrated the Gabor wavelet representation of face images with an enhanced discriminator, the Complete Kernel Fisher Discriminant (CKFD), and fractional power polynomial (FPP) models to improve recognition performance and discrimination ability. The continuous functions in the orthogonal moments are employed as kernel functions, as they are not affected by rotation, scaling, or translation. The orthogonal moments include Legendre moments [18,19], Zernike moments [20], Fourier–Mellin moments [21], Chebyshev–Fourier moments [22], and so on. Due to the low efficiency of traditional image recognition methods, scholars have studied the application of fractional moments. Zhang et al. [23] adopted fractional-order orthogonal Fourier–Mellin moments, which can improve the calculation performance of image moments by removing the factorial term in orthogonal polynomials. El Ogri et al. [24] used fractional generalized Laguerre moment invariants (FrGLMIs) to realize pattern recognition. Kaur et al. [25] used a support vector machine and fractional-order Zernike moments (FrZMs). Hosny et al. [26] created a set of fractional-order shifted Gegenbauer moments (FrSGMs) for image understanding and recognition. Horlando et al. [27] adopted fractional-order circular moments to solve some problems in image analysis. Guo et al. [28] introduced the Fractional-Order Fish Migration Optimization algorithm, which provides an optimal solution that can easily skip the whole order speed by using a new position generation strategy based on a global optimal solution. Zhang et al. [29] used fractional-order differentiation and closed image matting to perform multifocus image fusion. 

Three-dimensional images find extensive applications in various fields including medicine, industry, and the military. For many applications in these areas, efficient identification and accurate analysis are essential. However, due to the large amount of 3D image data, its high complexity, and the need to capture both local and global features, traditional methods usually face a series of challenges.

A novel 3D image recognition approach, integrating fractional Chebyshev moments with deep neural networks, offers a promising solution to the aforementioned challenges. In this method, fractional Chebyshev moments are combined with deep neural networks. The concept of fractional calculus is used to extract multiscale, nonuniform, and nonlocal information from 3D images. The deep neural networks are combined with global spatial information for feature fusion and classification recognition. This method can effectively streamline network parameter selection, enhancing both the accuracy and speed of 3D image recognition. The benefits of this method include the following:

Improved recognition accuracy: The traditional 3D image recognition model may be inaccurate due to unreasonable network design or insufficient extraction of data features. The fractional Chebyshev moment and deep neural network combined method can capture local and global features in 3D images more comprehensively and accurately and can improve recognition accuracy.

Reduced complexity: For large-scale 3D image data processing, the implementation of traditional methods requires significant human effort, material resources, and time costs. In this method, a fractional Chebyshev moment algorithm is introduced for multidimensional feature extraction, and a deep convolutional neural network (DCNN) is used to quickly and accurately classify the processed data, thus ensuring the accuracy and reducing the complexity of network parameter selection.

Significant practical value: The efficient 3D image recognition method combining fractional Chebyshev moments and deep neural networks has been widely used and has achieved good performance in image classification, face recognition, mapping and modeling, object recognition, and medical image recognition, especially when dealing with noise interference and other complex cases. It significantly contributes to improving image recognition performance.

Therefore, the efficient 3D image recognition method combining fractional Chebyshev moments and deep neural networks is of great significance, and it has a very wide application prospect in solving practical problems.

## 2. 3D Object Recognition Based on FrCMs and DNNs

### 2.1. Fractional-Order Chebyshev Moments

The FrCMs adopt successive integrals to compute Chebyshev moments and three recursive relation methods, which effectively achieve the invariants of rotation, translation, and scaling.

#### 2.1.1. Fractional-Order Chebyshev Moments

About the given function $f\left(x,y,z\right)$, the FrCMs of the order $\alpha \left(n+m+p\right)$ are defined within the region $\left[0,1\right]\times \left[0,1\right]\times \left[0,1\right]$, and it is possible to calculate this by continuous integration.
(1)$$FrCM_{nmp}^{\alpha }=\int_{0}^{1}\int_{0}^{1}\int_{0}^{1}f\left(x,y,z\right)\tilde{F}{\tilde{T}}_{n}^{\alpha_{x}}\left(x\right)\tilde{F}{\tilde{T}}_{m}^{\alpha_{y}}\left(y\right)\tilde{F}{\tilde{T}}_{p}^{\alpha_{z}}\left(z\right)dxdydz,$$
where $\alpha_{x}$, $\alpha_{y}$, and $\alpha_{z}$ > 0, and $\tilde{F}{\tilde{T}}_{n}^{\alpha_{x}}\left(x\right)$, $\tilde{F}{\tilde{T}}_{m}^{\alpha_{y}}\left(y\right)$, and $\tilde{F}{\tilde{T}}_{p}^{\alpha_{z}}\left(z\right)$ are representative fractional-order Chebyshev polynomials.

For a digital image intensity function $f\left(i,j,k\right)$ of size N×M×K, the $FrCM_{nmp}^{\alpha }$ is expressed as
(2)$$FrCM_{nmp}^{\alpha }=\sum_{i=0}^{N-1}\sum_{j=0}^{M-1}\sum_{k=0}^{K-1}f\left(i,j,k\right)\tilde{F}{\tilde{T}}_{n}^{\alpha_{x}}\left(x_{i}\right)\tilde{F}{\tilde{T}}_{m}^{\alpha_{y}}\left(y_{j}\right)\tilde{F}{\tilde{T}}_{p}^{\alpha_{z}}\left(z_{k}\right)\Delta_{x}\Delta_{y}\Delta_{z}$$

$\Delta_{x}=\frac{1}{N}$, $\Delta_{y}=\frac{1}{M}$, and $\Delta_{z}=\frac{1}{K}$, and the image coordinate of the mapping is expressed as
(3)$$x_{i}=\frac{i}{N}+\frac{\Delta_{x}}{2},y_{j}=\frac{j}{M}+\frac{\Delta_{y}}{2},z_{k}=\frac{k}{K}+\frac{\Delta_{z}}{2},$$
where i=1,2,⋯,N, j=1,2,⋯,M, and k=1,2,⋯K.

The original image is approximated as
(4)$$f\left(i,j,k\right)=\sum_{i=0}^{n_{max}}\sum_{j=0}^{m_{max}}\sum_{k=0}^{p_{max}}FrCM_{nmp}^{\alpha }\tilde{F}{\tilde{T}}_{n}^{\alpha_{x}}\left(x_{i}\right)\tilde{F}{\tilde{T}}_{m}^{\alpha_{y}}\left(y_{j}\right)\tilde{F}{\tilde{T}}_{p}^{\alpha_{z}}\left(z_{k}\right).$$

#### 2.1.2. Fractional-Order 3D Moment Invariants

Given the image function $f\left(i,j,k\right)$, 3D fractional-order moment invariants (FrGMs) of order $\left(\alpha_{x}p+\alpha_{y}q+\alpha_{z}r\right)$ defined continuously on the region N×M×K can be expressed as follows:(5)$$FrGM_{pqr}^{\alpha_{x}\alpha_{y}\alpha_{z}}=\sum_{i=0}^{N-1}\sum_{j=0}^{M-1}\sum_{0}^{K-1}f\left(i,j,k\right)m_{pqr}^{\alpha_{x}\alpha_{y}\alpha_{z}}\left(x_{i},y_{j},z_{k}\right),$$
where
(6)$$m_{pqr}^{\alpha_{x}\alpha_{y}\alpha_{z}}\left(x_{i},y_{j},z_{k}\right)=\int_{x_{i}-\frac{\Delta x_{i}}{2}}^{x_{i}+\frac{\Delta x_{i}}{2}}\int_{y_{j}-\frac{\Delta y_{j}}{2}}^{y_{j}+\frac{\Delta y_{j}}{2}}\int_{z_{k}-\frac{\Delta z_{k}}{2}}^{z_{k}+\frac{\Delta z_{k}}{2}}x^{\alpha_{x}p}y^{\alpha_{y}q}z^{\alpha_{z}r}dxdydz.$$By dividing the function, it simplifies to
(7)$$m_{pqr}^{\alpha_{x}\alpha_{y}\alpha_{z}}\left(x_{i},y_{j},z_{k}\right)=I_{Xp}^{\alpha_{x}}\left(x_{i}\right)I_{Yq}^{\alpha_{y}}\left(y_{j}\right)I_{Zr}^{\alpha_{z}}\left(z_{k}\right),$$
where
(8)$$\begin{array}{l} I_{Xp}^{\alpha_{x}}\left(x_{i}\right)=\int_{x_{i}-\frac{\Delta x_{i}}{2}}^{x_{i}+\frac{\Delta x_{i}}{2}}x^{\alpha_{x}p}dx=\frac{1}{\alpha_{x}p+1}\left[u_{i+1}^{\alpha_{x}p+1}-u_{i}^{\alpha_{x}p+1}\right]\\ I_{Yq}^{\alpha_{y}}\left(y_{j}\right)=\int_{y_{j}-\frac{\Delta y_{j}}{2}}^{y_{j}+\frac{\Delta y_{j}}{2}}y^{\alpha_{y}q}dy=\frac{1}{\alpha_{y}p+1}\left[\upsilon_{j+1}^{\alpha_{y}q+1}-\upsilon_{j}^{\alpha_{y}q+1}\right]\\ I_{Zr}^{\alpha_{z}}\left(z_{k}\right)=\int_{z_{k}-\frac{\Delta z_{k}}{2}}^{z_{k}+\frac{\Delta z_{k}}{2}}z^{\alpha_{z}r}dz=\frac{1}{\alpha_{z}r+1}\left[w_{k+1}^{\alpha_{z}r+1}-w_{k}^{\alpha_{z}r+1}\right] \end{array}.$$After merging, the expression is expressed as
(9)$$m_{pqr}^{\alpha_{x}\alpha_{y}\alpha_{z}}=\frac{1}{\left(\alpha_{x}p+1\right)\left(\alpha_{y}q+1\right)\left(\alpha_{z}r+1\right)}\left[u_{i+1}^{\alpha_{x}p+1}-u_{i}^{\alpha_{x}p+1}\right]\left[\upsilon_{j+1}^{\alpha_{y}q+1}-\upsilon_{j}^{\alpha_{y}q+1}\right]\left[w_{k+1}^{\alpha_{z}r+1}-w_{k}^{\alpha_{z}r+1}\right],$$
where
(10)$$\begin{array}{l} u_{i}=\left(i-0.5\right)\Delta x_{i};\upsilon_{j}=\left(j-0.5\right)\Delta y_{j};w_{k}=\left(k-0.5\right)\Delta z_{k}\\ u_{i+1}=\left(i+0.5\right)\Delta x_{i};\upsilon_{j+1}=\left(j+0.5\right)\Delta y_{j};w_{k+1}=\left(k+0.5\right)\Delta z_{k} \end{array}.$$The expression for the center point is denoted as
(11)$$X=\frac{FrGM_{100}^{\alpha_{x}\alpha_{y}\alpha_{z}}}{FrGM_{000}^{\alpha_{x}\alpha_{y}\alpha_{z}}},Y=\frac{FrGM_{010}^{\alpha_{x}\alpha_{y}\alpha_{z}}}{FrGM_{000}^{\alpha_{x}\alpha_{y}\alpha_{z}}},Z=\frac{FrGM_{001}^{\alpha_{x}\alpha_{y}\alpha_{z}}}{FrGM_{000}^{\alpha_{x}\alpha_{y}\alpha_{z}}}.$$The expression for the fractional-order translational invariant center distance is expressed as
(12)$$\eta_{pqr}^{\alpha_{x}\alpha_{y}\alpha_{z}}=\sum_{i=0}^{N-1}\sum_{j=0}^{M-1}\sum_{k=0}^{K-1}f\left(i,j,k\right)T_{pqr}^{\alpha_{x}\alpha_{y}\alpha_{z}}\left(x_{i},y_{j},z_{k}\right),$$
where
(13)$$T_{pqr}^{\alpha_{x}\alpha_{y}\alpha_{z}}=\int_{x_{i}-\frac{\Delta x_{i}}{2}}^{x_{i}+\frac{\Delta x_{i}}{2}}\int_{y_{j}-\frac{\Delta y_{j}}{2}}^{y_{j}+\frac{\Delta y_{j}}{2}}\int_{z_{k}-\frac{\Delta z_{k}}{2}}^{z_{k}+\frac{\Delta z_{k}}{2}}{\left(x-\hat{x}\right)}^{\alpha_{x}p}{\left(y-\hat{y}\right)}^{\alpha_{y}q}{\left(z-\hat{z}\right)}^{\alpha_{z}r}dxdydz.$$Based on the divisibility of the functional moments, Equation (10) simplifies to
(14)$$T_{pqr}^{\alpha_{x}\alpha_{y}\alpha_{z}}\left(x_{i},y_{j},z_{k}\right)=IT_{Xp}^{\alpha_{x}}\left(x_{i}\right)IT_{Yq}^{\alpha_{y}}\left(y_{j}\right)IT_{Zr}^{\alpha_{z}}\left(z_{k}\right),$$
where
(15)$$\begin{array}{l} IT_{Xp}^{\alpha_{x}}\left(x_{i}\right)=\frac{1}{\alpha_{x}p+1}\left[{\left(u_{i+1}-\hat{x}\right)}^{\alpha_{x}p+1}-{\left(u_{i}-\hat{x}\right)}^{\alpha_{x}p+1}\right]\\ IT_{Yq}^{\alpha_{y}}\left(y_{j}\right)=\frac{1}{\alpha_{y}q+1}\left[{\left(\upsilon_{j+1}-\hat{y}\right)}^{\alpha_{y}q+1}-{\left(\upsilon_{j}-\hat{y}\right)}^{\alpha_{y}q+1}\right]\\ IT_{Zr}^{\alpha_{z}}\left(z_{k}\right)=\frac{1}{\alpha_{z}r+1}\left[{\left(w_{k+1}-\hat{z}\right)}^{\alpha_{z}r+1}-{\left(w_{k}-\hat{z}\right)}^{\alpha_{z}r+1}\right] \end{array}.$$The merged expressions are organized as follows:(16)$$\begin{array}{l} T_{pqr}^{\alpha_{x}\alpha_{y}\alpha_{z}}=\frac{1}{\left(\alpha_{x}p+1\right)\left(\alpha_{y}q+1\right)\left(\alpha_{z}r+1\right)}\left[{\left(u_{i+1}-\hat{x}\right)}^{\alpha_{x}p+1}-{\left(u_{i}-\hat{x}\right)}^{\alpha_{x}p+1}\right]\\ \times \left[{\left(\upsilon_{j+1}-\hat{y}\right)}^{\alpha_{y}q+1}-{\left(\upsilon_{j}-\hat{y}\right)}^{\alpha_{y}q+1}\right]\left[{\left(w_{k+1}-\hat{z}\right)}^{\alpha_{z}r+1}-{\left(w_{k}-\hat{z}\right)}^{\alpha_{z}r+1}\right] \end{array}.$$

The 3D fractional-order moment invariants have rotational invariants, and the rotation of the matrix can be obtained as
(17)$$R_{xyz}\left(\theta ,\varphi ,\psi \right)=\left(\begin{array}{ccc} \cos\varphi \cos\psi & \cos\varphi \sin\psi & -\sin\varphi \\ \sin\theta \sin\varphi \cos\psi -\cos\theta \sin\psi & \sin\theta \sin\varphi \sin\psi +\cos\theta \cos\psi & \cos\varphi \sin\theta \\ \cos\theta \sin\varphi \cos\psi +\sin\theta \sin\psi & \cos\theta \sin\varphi \sin\psi -\sin\theta \cos\psi & \cos\theta \cos\varphi \end{array}\right).$$

The rotation matrix is usually used for the linear transformation of the object coordinates, as shown below:(18)$$\left(\begin{array}{c} x^{\prime }\\ y^{\prime }\\ z^{\prime } \end{array}\right)=R_{xyz}\left(\theta ,\varphi ,\psi \right)\left(\begin{array}{c} x-\hat{x}\\ y-\hat{y}\\ z-\hat{z} \end{array}\right)=\left(\begin{array}{ccc} R_{11} & R_{12} & R_{13}\\ R_{21} & R_{22} & R_{23}\\ R_{31} & R_{32} & R_{33} \end{array}\right)\left(\begin{array}{c} x-\hat{x}\\ y-\hat{y}\\ z-\hat{z} \end{array}\right),$$
where ${\left(R_{ij}\right)}_{\begin{array}{l} 1\leq i\leq m\\ 1\leq j\leq n \end{array}}$ is an element of the matrix $R_{xyz}\left(\theta ,\varphi ,\psi \right)$.

The 3D moment invariants of fractional order are simply written as FrGMIs:(19)$$FrGMI_{pqr}^{\alpha_{x}\alpha_{y}\alpha_{z}}=\lambda^{-\gamma }\sum_{i=0}^{N-1}\sum_{j=0}^{M-1}\sum_{k=0}^{K-1}f\left(i,j,k\right)\mu_{pqr}^{\alpha_{x}\alpha_{y}\alpha_{z}}\left(x_{i},y_{j},z_{k}\right),$$
where
(20)$$\mu_{pqr}^{\alpha_{x}\alpha_{y}\alpha_{z}}=\int_{x_{i}-\frac{\Delta x_{i}}{2}}^{x_{i}+\frac{\Delta x_{i}}{2}}\int_{y_{j}-\frac{\Delta y_{j}}{2}}^{y_{j}+\frac{\Delta y_{j}}{2}}\int_{z_{k}-\frac{\Delta z_{k}}{2}}^{z_{k}+\frac{\Delta z_{k}}{2}}\left[\begin{array}{c} {\left(R_{11}\left(x-\hat{x}\right)+R_{12}\left(y-\hat{y}\right)+R_{13}\left(z-\hat{z}\right)\right)}^{\alpha_{x}p}\\ {\left(R_{21}\left(x-\hat{x}\right)+R_{22}\left(y-\hat{y}\right)+R_{23}\left(z-\hat{z}\right)\right)}^{\alpha_{y}q}\\ {\left(R_{31}\left(x-\hat{x}\right)+R_{32}\left(y-\hat{y}\right)+R_{33}\left(z-\hat{z}\right)\right)}^{\alpha_{z}r} \end{array}\right]dxdydz.$$

The normalized parameters are expressed as
(21)$$\begin{array}{l} \lambda =FrGM_{000}^{\alpha_{x}\alpha_{y}\alpha_{z}},\gamma =1+\frac{\alpha_{x}n+\alpha_{y}m+\alpha_{z}p}{3}\\ \theta =\frac{1}{2}tan^{-1}\left(\frac{2\eta_{011}}{\eta_{020}+\eta_{002}}\right),\phi =\frac{1}{2}tan^{-1}\left(\frac{2\eta_{101}}{\eta_{200}+\eta_{002}}\right)\\ \psi =\frac{1}{2}tan^{-1}\left(\frac{2\eta_{110}}{\eta_{200}+\eta_{020}}\right) \end{array}.$$

The exact calculation of fractional-order 3D moment invariants is not possible and can only be approximated, but due to the divisibility of the function moments, it is difficult to reduce the triple integral $\mu_{pqr}^{\alpha_{x}\alpha_{y}\alpha_{z}}$ to a simple integral using the numerical integration algorithm and the 3D Gaussian integration method.

Three-dimensional Gaussian integration is a way to integrate functions in three-dimensional space by dividing the three-dimensional space into infinitely small volume elements and accumulating the values of functions within the volume element multiplied by the volume of that volume element.

Gaussian integration is capable of solving various types of orthogonal polynomials, including Legendre, Chebyshev, Laguerre, Hermite, and others. However, there are still computational errors, and the integration interval should not be too large, otherwise the results will be inaccurate. In addition, the error of Gaussian integration is also related to the smoothness of the product function, and the worse the smoothness of the product function, the larger the error. Therefore, the Gaussian integral is the formula with the highest algebraic accuracy for a given number of nodes.

It can enhance the computational efficiency of fractional-order 3D moment invariants under the condition that the fractional parameters are satisfied, and it can accurately perform the calculation and achieve a desirable result for different types of 3D images, regardless of whether they are processed by rotation, scaling, translation transformation, noise processing, or filtering.

#### 2.1.3. Fractional Chebyshev Moment Invariants

Fractional order can perform image recognition of 3D objects, and common moments can only achieve rotational transformation. Since Chebyshev moments add the two transformations of scaling and translation, fractional order is fused with Chebyshev moments to form fractional-order Chebyshev moments. By normalizing the transformed 3D objects and obtaining the rotational invariants of the parameters, the computing efficiency is improved under the premise of satisfying the parameter accuracy to achieve high efficiency in 3D image recognition.

The digital image strength function $f\left(i,j,k\right)$ is weighted to obtain the weighted image intensity function $\tilde{f}\left(i,j,k\right)$. FrCMs can be described as
(22)$$\begin{array}{l} FrCM_{nmp}^{\alpha }=\int_{0}^{1}\int_{0}^{1}\int_{0}^{1}\tilde{f}\left(i,j,k\right)\tilde{F}{\tilde{T}}_{n}^{\alpha_{x}}\left(x_{i}\right)\tilde{F}{\tilde{T}}_{m}^{\alpha_{y}}\left(y_{j}\right)\tilde{F}{\tilde{T}}_{p}^{\alpha_{z}}\left(z_{k}\right)dxdydz\\ =\frac{1}{\sqrt{d_{n,\alpha_{x}}^{2}d_{m,\alpha_{y}}^{2}d_{p,\alpha_{z}}^{2}}}\int_{0}^{1}\int_{0}^{1}\int_{0}^{1}f\left(i,j,k\right)\tilde{F}{\tilde{T}}_{n}^{\alpha_{x}}\left(x_{i}\right)\tilde{F}{\tilde{T}}_{m}^{\alpha_{y}}\left(y_{j}\right)\tilde{F}{\tilde{T}}_{p}^{\alpha_{z}}\left(z_{k}\right)dxdydz \end{array},$$
where
(23)$$\tilde{f}\left(i,j,k\right)={\left\lfloor \begin{array}{ccc} w_{\alpha_{x}}\left(x\right) & w_{\alpha_{y}}\left(y\right) & w_{\alpha_{z}}\left(z\right) \end{array}\right\rfloor }^{-1/2}f\left(i,j,k\right).$$
(24)$$FrCM_{nmp}^{\alpha }=\frac{1}{\sqrt{d_{n,\alpha_{x}}^{2}d_{m,\alpha_{y}}^{2}d_{p,\alpha_{z}}^{2}}}\sum_{l=0}^{n}\sum_{s=0}^{m}\sum_{r=0}^{p}B_{n,l}B_{m,s}B_{p,r}FrGM_{pqr}^{\alpha_{x}\alpha_{y}\alpha_{z}},$$
where $FrGM_{pqr}^{\alpha_{x}\alpha_{y}\alpha_{z}}$ is the fractional geometric moment.

Chebyshev moment invariants can be denoted as
(25)$$FrCMI_{nmp}^{\alpha }=\frac{1}{\sqrt{d_{n,\alpha_{x}}^{2}d_{m,\alpha_{y}}^{2}d_{p,\alpha_{z}}^{2}}}\sum_{l=0}^{n}\sum_{s=0}^{m}\sum_{r=0}^{p}B_{n,l}B_{m,s}B_{p,r}FrGMI_{pqr}^{\alpha_{x}\alpha_{y}\alpha_{z}},$$
where the denominators of $\alpha_{x}$, $\alpha_{y}$, and $\alpha_{z}$ are odd.

### 2.2. FrCMs-DNNs Model

DNNs, the multilayer unsupervised neural networks, systematically map features layer by layer to acquire an improved representation of the input. These networks incorporate a range of nonlinear mapping feature transformations for handling highly complex functions. Viewing the deep structure as a neuronal network, the core idea of a deep neural network can be succinctly described as follows:

(1) Pre-training the network with unsupervised learning methods;

(2) Layer-by-layer training using unsupervised learning;

(3) Fine-tuning the network model with supervised learning.

DNNs are constructed upon the foundational perceptron model, which is a multiple-input single-output model. This model learns a linear relationship between inputs and outputs to generate the desired outputs.
(26)$$z=\sum_{i=1}^{m}w_{i}x_{i}+b,$$
where w is the parameter of weight coefficient and b is the bias amount.

The output result is obtained following the neuronal activation function:(27)$$sign\left(z\right)=\left\{\begin{array}{l} -1,z<0\\ 1,z\geq 0 \end{array}\right..$$

Although the structure of DNNs is very complex, its essence is still a perceptual system. The algorithm starts from any input layer, runs from left to right, and obtains a result at the last output layer. When the calculation results deviate greatly from the target value, the errors of each node are inverted from right to left, and the total weight of each node is modified. After reaching the input layer in reverse, the operation continues and repeats until each weight reaches an appropriate value. Compared with the traditional mathematical analysis, some parameters of this kind of differential equation adopt the mode of random selection and finally make it more accurate by modifying it.

The algorithm can avoid the prior knowledge of each level of information and improve its performance. At the same time, its quantitative level enables the algorithm to deeply learn the distributed information and improve its effectiveness. Compared with the shallow model, the depth model can better describe the real information, with stronger details and better description ability, so that it can better identify the image effectively.

As the same with the architecture of DNNs, the structure of FrCMs-DNNs contains three layers, an input, hidden layers, and an output that are fully connected. By adjusting the weights and biases, FrCMs-DNNs achieves an output with the expected accuracy relative to the network input. Table 1 contains a detailed description of the FrCMs-DNNs system.

As depicted in Figure 1, the FrCMs-DNNs model is computationally efficient and has a small memory requirement, and it is suitable for use in the ABC optimizer algorithm [30] for problems with a large amount of data and parameters.

The FrCMs-DNNs model includes an input, hidden layers, and an output. The output from the softmax layer corresponds to the number of classification labels. The input is expressed in terms of 3D FrCMs, and the descriptor vector consists of order r of 3D FrCMs, with r set by the experiment. The input vector can be expressed as
(28)$$V=\left[FrCM_{nmp}^{\alpha }\left|n\times m\times p\in \left[0,1,\cdots ,r\right]\right.\right].$$

When the maximum order of 20 × 20 × 20 is used, V denotes 8000 dimensions. The dataset is categorized into two parts, the training and test sets. The hidden layers are used in the model with four hidden layers, containing 100, 165, 240, and 120 neurons.
(29)$$Y_{i}=\eta_{i}\left(b^{\left(i\right)}+W^{\left(i\right)}Y_{i}\right).$$

$Y_{i}$ represents the output of the hidden layer i, $W^{\left(i\right)}$ denotes the weight coefficient matrix, $b^{\left(i\right)}$ denotes the bias vector, and $\eta_{i}$ denotes the activation function.

The softmax function is an extension of the logic function:(30)$$softmax\left(y^{j}\right)=\left(\frac{e^{y^{j}}}{\sum_{i=1}^{s}e^{y^{i}}}\right),j=0,1,\cdots ,s.$$

The output of the model can be calculated as
(31)$$f\left(V\right)=softmax\left(b^{\left(5\right)}+W^{\left(5\right)}Y_{4}\right).$$

BN: Batch normalization improves the learning rate, accelerates training, and avoids divergence and overfitting.

ELU: The exponential linear unit brings the average value of the activation function close to zero to speed up the learning. It enables avoidance of the problem of gradient disappearance.

ReLU: The rectified linear unit can be defined as $f=\max\left(0,x\right)$; it is insensitive to the gradient vanishing problem and improves the convergence speed.

Softmax function: The softmax function compresses a vector z of real numbers into another real vector $\sigma \left(z\right)$, ensuring that each element falls within the range $\left(0,1\right)$ and the sum of the elements is 1.

Fractional-order Chebyshev moments and DNNs for 3D image recognition are shown in Table 2, including their main aspects, common datasets, characteristics, evaluation index, advantages, and limitations.

## 3. Experiment

In this section, we designed three experiments from the perspectives of the effectiveness of FrCMs, 3D recognition ability, and practical application value, respectively. Firstly, fractional moments and 3D reconstruction play an important role in image feature tasks, and the relationship between them reflects the model’s ability to retain and reconstruct image information. Therefore, image reconstruction experiments are designed to evaluate the reconstruction results through MSE and PSNR indicators and analyze the evaluation model’s ability to extract image information effectively. Secondly, to prove that FrCMs-DNNs has certain advantages in multiscale feature extraction, global feature learning, robustness, and generalization ability in 3D recognition tasks, comparison and ablation experiments for FrCMs-DNNs are designed, and different evaluation indicators are used to quantitatively analyze the effectiveness of the method. Finally, to prove that FrCMs-DNNs is good at extraction and feature representation in image recognition tasks and has strong adaptability and expression ability, the recognition experiment based on SAR image is designed to analyze and verify the universality and robustness of the method.

### 3.1. 3D Image Reconstruction

Experiment 1: Image reconstruction was carried out with different fractional-order moments.

Experiment 2: Different moment order parameters were used for feature extraction.

In this paper, MSE and PSNR, commonly used evaluation indexes, are used to compare the quality of images reconstructed with different fractional moments.

For the image whose original image $f\left(x,y,z\right)$ and reconstructed image $\hat{f}\left(x,y,z\right)$ size are both N×M×K, the mean square error is defined as:(32)$$MSE=\frac{1}{NMK}{\sum_{x=0}^{N-1}\sum_{y=0}^{M-1}\sum_{z=0}^{K-1}\left\|f\left(x,y,z\right)-\hat{f}\left(x,y,z\right)\right\|}^{2}.$$

The MSE is the average of the differences between the two images, but as the quantization number increases, the MSE becomes larger. Hence, a smaller MSE value indicates better image reconstruction quality.

PSNR, defined by MSE, is frequently employed as a metric for signal reconstruction quality, particularly in fields like image compression. When the gray level of the image is set as L(8-bit gray level image L is 255), then
(33)$$PSNR=10\cdot \mathrm{lg}\frac{L^{2}}{MSE}.$$

The value range of *PSNR* is $\left(0,+\infty \right)$. A larger value of PSNR means a better performance.

According to Figure 2, when the order of the matrix is 40, most of the fractional-order moments have a good reconstruction effect in 3D ant images. And as shown in Figure 3, with the increase in order, the MSE values of FrCMs, FrOLMs, and FrGLMs gradually approach 0, and their PSNR values gradually increase. When the order reaches the highest order of 150, the PSNR values of FrOFMMs [38] and FrZMs are 0, the PSNR value of FrCMs is 44, the PSNR value of FrOLMs is 37, and the PSNR value of FrCMs is 31.

Image reconstruction results are applied to evaluate the performance of the presented method. Figure 2 and Figure 3 show the comparison of the image reconstruction results of FrCMs, FrOFMMs, FrZMs, FrOLMs, and FrGLMs. With increasing fractional-order moments, the effect of the reconstructed image approaches that of the raw image. The reconstruction result of FrCMs is the best when the parameters are chosen as $\alpha_{x}=1.2$, $\alpha_{y}=1$, and $\alpha_{z}=1.2$, which verifies that the fractional Chebyshev moments can effectively achieve visual reconstruction.

### 3.2. Feature Extraction

The image reconstruction was performed using the “bird” 3D image dataset, which is a combination of the original bird images. Multiple reconstruction experiments of bird images using different fractional parameters were conducted, which can verify the local image extraction ability of FrCMs.

Figure 4 shows the feature extraction results of 3D images by FrCMs with different $\alpha_{x}$, $\alpha_{y}$, $\alpha_{z}$, and maximum orders. The FrCMs accurately represent the image information and show the ability of efficient local feature extraction. The approximate error of the FrCMs is smaller than the existing moments, and the advantages are obvious in 3D image reconstruction.

### 3.3. 3D Object Recognition

The PSB dataset [39] and the medical images dataset [40] are used to validate the effectiveness of 3D image recognition.

Experiment 1: As shown in Figure 5, 20 objects of different categories are selected in the PSB dataset, and as shown in Figure 6, 12 objects of different categories are selected in the medical images dataset, which can verify the recognition capability of FrCMIs (fractional-order Chebyshev moment invariants), respectively.

The noise robustness of the FrCMIs is tested by rotating, scaling, and translating the dataset objects with transformations. The classification accuracy of the FrCMIs is evaluated by adding various densities of Gaussian noise. The performance of FrCMIs in 3D object classification is compared with that of FrFMMIs, FrZMIs, FrLMIs, and GMIs. The fractional parameters of FrCMIs are set up as follows:$$\begin{array}{c} (1)\;\alpha_{x}=1.4,\alpha_{y}=1.4,\alpha_{z}=1.4;\;(2)\;\alpha_{x}=1.4,\alpha_{y}=1.0,\alpha_{z}=0.8;\\ (3)\;\alpha_{x}=0.8,\alpha_{y}=1.4,\alpha_{z}=1.0;\;(4)\;\alpha_{x}=0.8,\alpha_{y}=0.8,\alpha_{z}=0.8. \end{array}$$

Fractional-order moment invariants are used to process 3D objects with different densities of Gaussian noise on the PSB dataset and medical image dataset. Gaussian noise is caused by the random noise of an image sensor. It is random and follows Gaussian distribution. It will make the brightness and color of the image have slight random changes, as well as cause blurring and distortion.

In the image processing experiments, the Gaussian noise density is chosen to be at most 10%, and according to the previous experimental rules, Gaussian noise with a density of 1–5% will be selected for experimental verification, as shown in Figure 7 and Figure 8, for the PSB dataset and medical dataset, with different densities of Gaussian noise processing for 3D object recognition.

Table 3 and Table 4 depict the object recognition rate results for the PSB dataset and the medical images dataset, respectively. The description of the followed methods is shown in Appendix A. 

As shown in Table 3 and Table 4, there is a difference between the object recognition rate presented by adding Gaussian noise and no noise in the 3D image, and the recognition effect of the 3D object treated with no noise is close to the exact recognition compared with the 3D object treated with Gaussian noise; according to the different densities of the 3D object treated with Gaussian noise, the higher the density, the greater the error caused by the recognition of the 3D object in the recognition process. The higher the density, the greater the error caused in the recognition process, resulting in a lower recognition rate in the end; by comparing different types of fractional-order moment invariants in the recognition process of 3D objects without noise and Gaussian noise, it can be seen that the proposed FrCMIs recognition effect is the best, and the recognition rate is the highest among all fractional-order moment invariants.

Experiment 2: The datasets are constructed by performing a series of transformations on selected objects. The PSB dataset includes 10 categories: airplane, ant, bird, cup, fish, hand, octopus, spider, glasses, and teddy bear. The medical images dataset includes 5 categories: head, abdomen, hip, knee, and leg. In the classification task of the PSB dataset, 240 (40%) objects are randomly selected as the training set, and 360 (60%) objects are selected as the test set. In the classification task of the medical images dataset, 150 (50%) objects are randomly selected as the training set, and 150 (50%) objects are selected as the test set.

As depicted in Figure 9, the recognition accuracy of FrCMs-DNNs is higher than FrCMIs, and the FrCMs-DNNs model has the best classification results. Figure 10 presents the confusion matrix of the fractional Chebyshev moments models for the PSB dataset [35] and the medical images dataset [36]. Most of the confused objects are nearly completely recognized, with a little amount of confusion between the bird/airplane and ant/octopus categories in the PSB dataset, and a small amount of confusion between the abdomen/ hip and knee/leg categories in the medical images dataset, since these categories have similar shapes.

The FrOLMs, FrOFMMs, FrZMs, FrCMs, and FrGLMs are adopted as input layers to demonstrate the classification capabilities of the FrCMs unit, respectively. The accuracy of corresponding classification results is obtained by adding the order of different fractional-order moments. As shown in Figure 11, compared with other methods combining fractional-order moments and DNNs, FrCMs-DNNs has the highest object recognition accuracy.

### 3.4. Ablation Experiment

The FrCMs-DNNs model consists of FrCMs and DNNs. In order to demonstrate the effectiveness of each module of the FrCMs-DNNs, the accuracy of FrCMs-DNNs was verified by ablation experiments.

In this experiment, the performance of the FrCMs-DNNs model applied to 3D object recognition was verified. In this experiment, the FrCM and DNN models used for 3D object recognition were taken as the benchmark model, and the PSB dataset and medical image dataset were chosen as the experimental datasets to verify the accuracy and universality of the FrCMs-DNNs model.

#### 3.4.1. Evaluation Methods and Indicators

In order to avoid the phenomenon of high accuracy due to the presence of all normal prediction sample data in the test data, $A_{Accuracy}$, precision rate $P_{\mathrm{Pr}ecision}$, recall rate RRecall, and F_1_-score are selected in the experiment. Here are the calculation formulae:(34)$$A_{Accuracy}=\frac{T}{N_{ALL}},$$
(35)$$P_{\mathrm{Pr}ecision}=\frac{T_{p}}{T_{P}+F_{p}},$$
(36)RRecall=TpTp+FN,
(37)F1=2×RRecall×PPrecisionRRecall+PPrecision,
where $N_{ALL}$ is the total sample number; T is the number of samples that are predicted correctly; $T_{P}$ is the normal and predicted normal sample number; $F_{P}$ is the number of samples where normal prediction is abnormal; and $F_{N}$ is the number of samples where the abnormal prediction is normal.

The accuracy value can directly reflect the overall accuracy of the method. The precision rate and recall rate can reflect whether the model is in the overfitting state. A low accuracy rate indicates that the model is biased to output abnormal labels, while a low recall rate indicates that the model is biased to output normal labels. The F_1_-score comprehensively reflects the precision rate and recall rate, and the higher the F_1_-score is, the better the model fitting effect is.

#### 3.4.2. Ablation Experiment and Analysis

As is shown in Figure 12, Experiment 1 compared the detection effect of the FrCM, DNN, and FrCMs-DNNs models based on the PSB dataset, set the same parameters for their models, and compared the 3D object recognition to assess the performance of FrCMs-DNNs. In the process of training, the method of a 5-fold crossover experiment was used to verify the results.

As is shown in Figure 13, Experiment 2 compared the universality of the FrCM, DNN, and FrCMs-DNNs models based on the medical image dataset, set the same parameters of their models, and used the undersampling method to avoid the imbalanced data phenomenon. The dataset was also trained by the method of five-fold cross-validation.

In summary, according to the results of Experiment 1, with the same parameters, the accuracy of FrCMs-DNNs is improved by 0.07 and 0.05, respectively, compared with FrCMs and DNNs. Its precision rate, recall rate, and F1-score are also among the leading levels. Note that FRCMs-DNNs takes into account the advantages of both FrCM and DNN models. Compared with a single model, the performance is significantly improved.

According to the results of Experiment 2, in the model migration experiment from the PSB dataset to medical image dataset, the FrCMs-DNNs model still maintains a high performance advantage, and the accuracy and F1-score are 0.04 and 0.01 higher than the FrCMs model, respectively. It can be concluded that the FrCMs-DNNs model has high accuracy.

In Experiment 3, to evaluate the computational efficiency of FrCMs-DNNs, the model will be validated for computational efficiency on CPUs and GPUs using the PSB dataset and the medical images dataset, and the computer configurations chosen are RTX 2080Ti GPUs and an Intel(R) Xeon Silver 4112 CPU@2.60GHz. To fully evaluate the performance of the approach in a different hardware environment, the computational efficiency of FrCMs is evaluated by floating point operations per second (FLOPs), training or inference speed (frame per second, FPS), and Top-1/Top-5 accuracy (%), as shown in Table 5 and Table 6.

As can be seen from Table 5 and Table 6, FrCMs-DNNs was implemented on different performance hardware for 3D image recognition with the same network parameters. The experimental results demonstrate that on GPU, the FLOPS value is approximately twice that of CPU, and the speed and accuracy of GPU training is better compared with CPU, which improves the computational efficiency more effectively; thus, the FrCMs-DNNs model can be better implemented in specific hardware environments. Therefore, for 3D image recognition, the FrCMs-DNNs model has better computational efficiency.

### 3.5. SAR Image Recognition

The feasibility of the FrCMs-DNNs model with high speed and accurate recognition is verified in the SAR image classification and detection experiments.

#### 3.5.1. SAR Image Ship Classification

Synthetic aperture radar (SAR) is a remote sensing technology that uses radar signals and signal processing techniques to create high-resolution radar images. With synthetic aperture technology, SAR systems can achieve very-high-resolution imaging that can provide detail-rich images of the target and obtain high-quality images.

FrCMs-DNNs is applied to SAR images using the public VAIS ship dataset [41], as shown in Figure 14, which has 1088 images, mainly including 6 coarse-grained categories, 5 categories of which are selected as merchant ships, medium passenger ships, sailing ships, small boats, and tugboats. In this paper, 477 images are randomly chosen for training, while 473 images are designated for testing.

In this experiment, to validate the performance of FrCMs-DNNs, which can be applied in image classification, fractional-order Chebyshev moments and deep neural network features are fused for SAR ship classification. FrCMs-DNNs is used for SAR image ship classification on the VAIS ship dataset, and the feasibility and robustness of its model are verified by commonly used evaluation methods and metrics.

In order to avoid the phenomenon that the prediction sample data in the test data are all normal and the accuracy is too high, the above-mentioned accuracy A_Accuracy_, inspection accuracy P_Precision_, recall R_Recall_, and F1-score are selected for the experiment. The results are presented in Table 7.

As evident from the table, FrCMs-DNNs is effective and robust for image recognition in SAR image ship classification using the VAIS ship dataset for the classification test, and the results show that the accuracy is above 80% and the model fits well when using the FrCMs-DNNs model as a feature vector to overcome the sensitivity of SAR images to orientation and effectively improve SAR image ship classification.

#### 3.5.2. High-Speed SAR Image Ship Detection

To validate the effectiveness of the proposed FrCMs-DNNs, high-resolution SAR images are employed for comparison. The model is compared with the mask attention interaction and scale enhancement network, grid convolutional neural network, and depthwise separable convolutional neural network.

The experimental dataset is all the 3000 SAR images containing ships within the area of Dalian Port, as required for high-speed SAR ship detection, including high-speed SAR ship images and corresponding labels (ship’s position, bounding box, etc.). Then, the dataset is pre-processed, including the operations of image denoising, normalization, cropping, etc., to ensure the quality and consistency of the input data. The effectiveness of the model is measured by evaluating the methods and metrics: accuracy A_Accuracy_, accuracy check P_Precision_, recall R_Recall_, and F1-score.

From Figure 15 and Figure 16, it can be seen that the high-speed SAR ship detection results of FrCMs-DNNs are better than those of the Mask attention interaction and scale enhancement networks; the grid convolutional neural network and the depthwise separable convolutional neural network have better detection results, and their accuracy rates are higher than those of the other three networks; and the proposed FrCMs-DNNs has good effectiveness.

The performance of the proposed FrCMs-DNNs is evaluated in image reconstruction and image classification experiments.

### 3.6. 3D Recognition Consumes Time

The efficiency of 3D recognition is also a matter of concern in the application. Being efficient and fast is also of great importance. In this section, different types of fractional moment DNN models are used to verify that the proposed FrCMs-DNNs model takes less time and recognizes well in the 3D recognition process.

The time consumption method for 3D recognition is influenced by many factors, such as the feature extraction method, classifier, and hardware conditions. In this section, the hardware condition method is used to calculate the time consumption. However, the time consumption is related to the configuration of the device, such as the GPU, CPU model, and memory size, which will affect the efficiency of 3D image recognition.

When running 3D object recognition experiments on a computer with performance parameters of AMD A10-7300 Radeon 610 Compute Cores 4C+6G, the elapsed time is recorded through the timeliness of the sensor as the order increases. The beginning and end times should be recorded for data processing, and the experiment should be repeated several times to avoid large errors.

In this experiment, 3D object recognition is performed using fractional-order moments–DNNs models of different orders with the fractional parameter α=1.4 under the PSB dataset of size 128×128×128 and the medical images dataset, respectively, and the time used for the fractional-order moments–DNNs model consumption corresponding to their recognition process is collected, as shown in Table 8 and Table 9.

As shown in Table 8 and Table 9 and Figure 17, it is evident that the time consumed by the fractional-order moments–DNNs model for recognizing 3D objects increases linearly with the gradual increase in moment order; the average time consumed by the proposed FrCMs-DNNs model is the shortest compared to other fractional-order moments–DNNs models, which can indicate that the FrCMs-DNNs model is efficient in 3D object recognition.

As a nonlinear feature extraction method, FrCMs can capture higher-order statistical information in data, but it is not easy to extract results from 3D images. However, deep neural networks have nonlinear activation functions, which can improve the extraction of complex structures and better adapt to the nonlinear data relationship. Utilizing fractional moments, FrCMs can reduce feature dimensions while preserving crucial information in the data. This aids in reducing the parameter count of the neural network model, lessening the computational burden of training and inference, and enhancing overall computational efficiency.

Therefore, the combination of fractional Chebyshev moments and deep neural networks can fully leverage the advantages of both, improve the extraction ability, adaptability, and robustness of FrCMs-DNNs, and reduce the dimensional requirements, making it more suitable for dealing with a variety of practical problems.

## 4. Limitations and Future Work

Fusing fractional Chebyshev moments and deep neural networks is a new image recognition technology, and its main characteristics include the following: (1)High efficiency: the method can recognize 3D images quickly and accurately, and the processing speed is fast;(2)High accuracy: the method combines the benefits of fractional Chebyshev moments and deep neural networks, effectively enhancing the accuracy of 3D image recognition;(3)High reliability: the method adopts the integration of multiple technologies to enhance the reliability of 3D image recognition and reduce misjudgment rates.

This method also has its limitations. It needs a lot of training data to enhance the recognition performance, and the quality of the training data is also high. High computing power and storage space are required to support model training and inference. It is necessary to set and optimize multiple sets of parameters, which requires a high technical level.

The development of image recognition algorithms will continue to explore the improvement of deep learning, multimodal recognition, and migration learning with fewer samples to enhance the accuracy, generalization, and adaptability of image recognition. Also, attempts can be made to combine fractional-order moments with other types of classifiers to build new image recognition algorithms.

## 5. Conclusions

The FrCMs-DNNs method for 3D image recognition is proposed by combining fractional Chebyshev moments and deep neural networks. The experimental results of 3D image reconstruction show that FrCMs have the smallest MSE and the highest accuracy of image reconstruction in comparison with other fractional moments. For the PSB dataset, the recognition accuracy of FrCMIs is 32.1% higher than the mean of other fractional-order moment invariants. For the medical images dataset, the recognition accuracy of FrCMIs is 27.3% higher than the mean of other fractional-order moment invariants. The recognition rate of the FrCMs-DNNs model surpasses that of other fractional-order moment–DNN models in the PSB and medical images datasets under the same parameters, and the average value of time consumed by the FrCMs-DNNs model for 3D object recognition in the PSB and medical images datasets is the smallest for different types of fractional-order moments–DNNs models under the condition of increasing order.

## Figures and Tables

**Figure 1 sensors-24-02352-f001:**
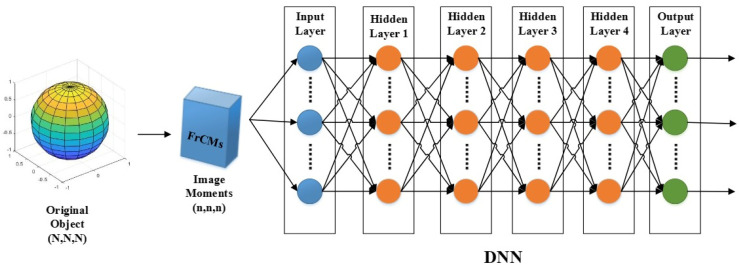
The model structure of FrCMs-DNNs for 3D object classification.

**Figure 2 sensors-24-02352-f002:**
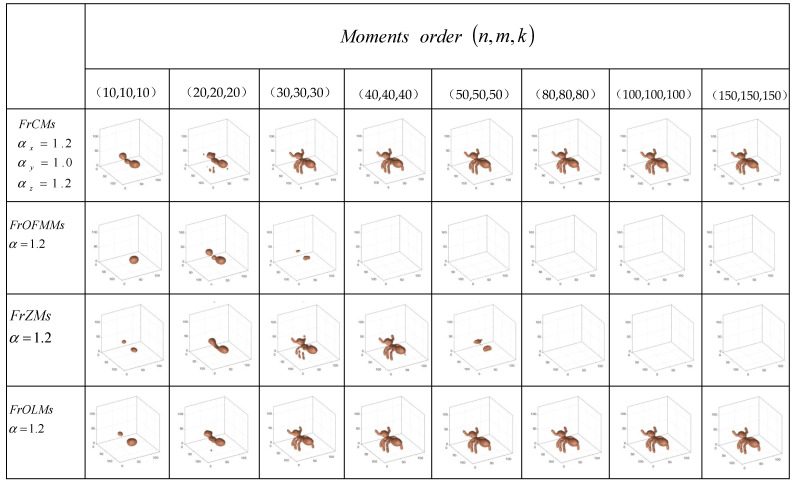
Comparison of various 3D image reconstruction results.

**Figure 3 sensors-24-02352-f003:**
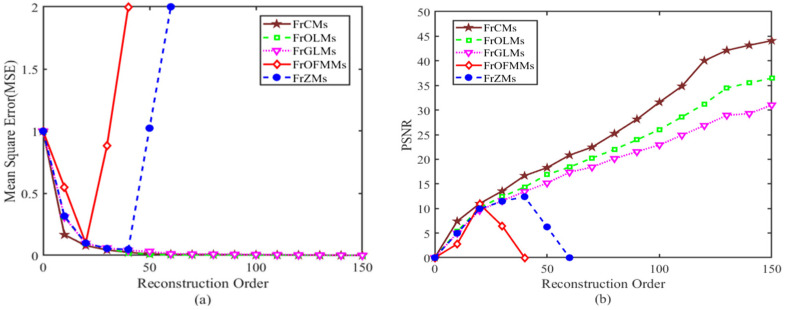
Comparison curves of different reconstructed methods from 3D image: (**a**) MSE; (**b**) PSNR.

**Figure 4 sensors-24-02352-f004:**
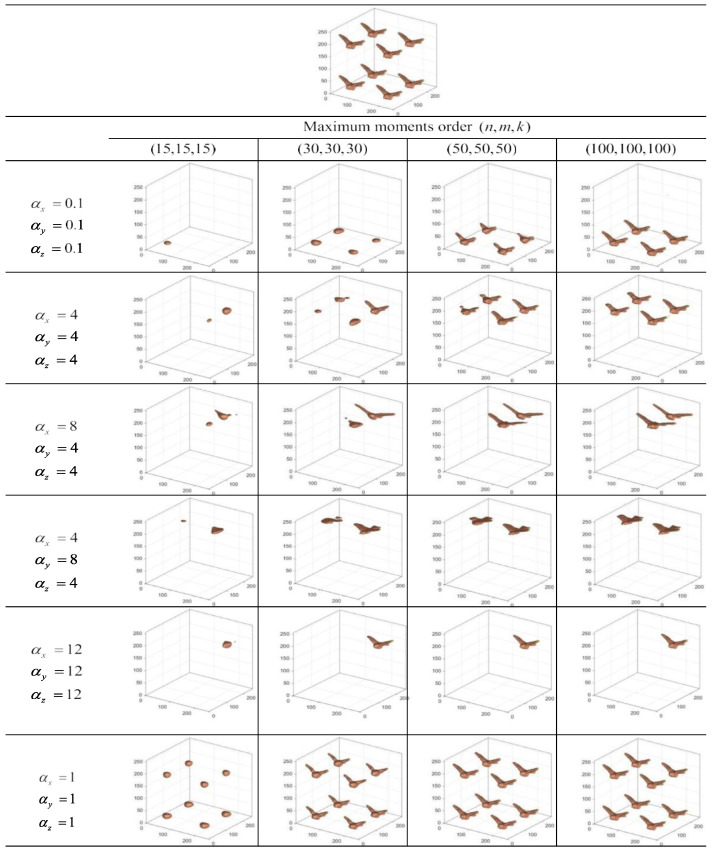
The local feature extraction results of FrCMs.

**Figure 5 sensors-24-02352-f005:**
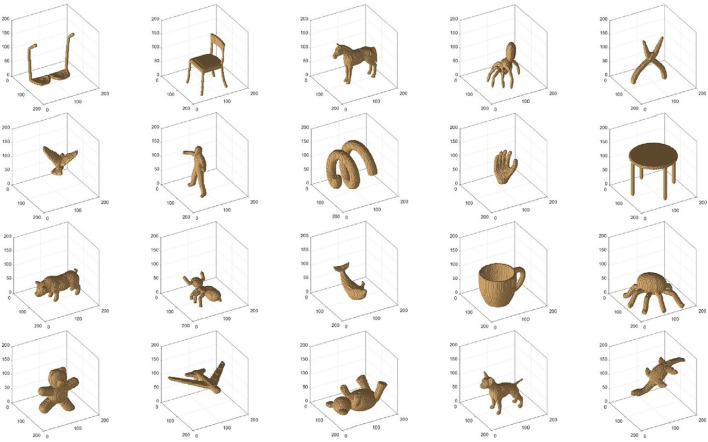
The PSB dataset.

**Figure 6 sensors-24-02352-f006:**
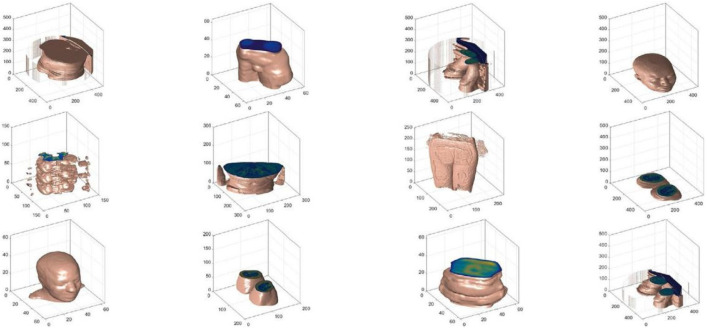
The medical images dataset.

**Figure 7 sensors-24-02352-f007:**
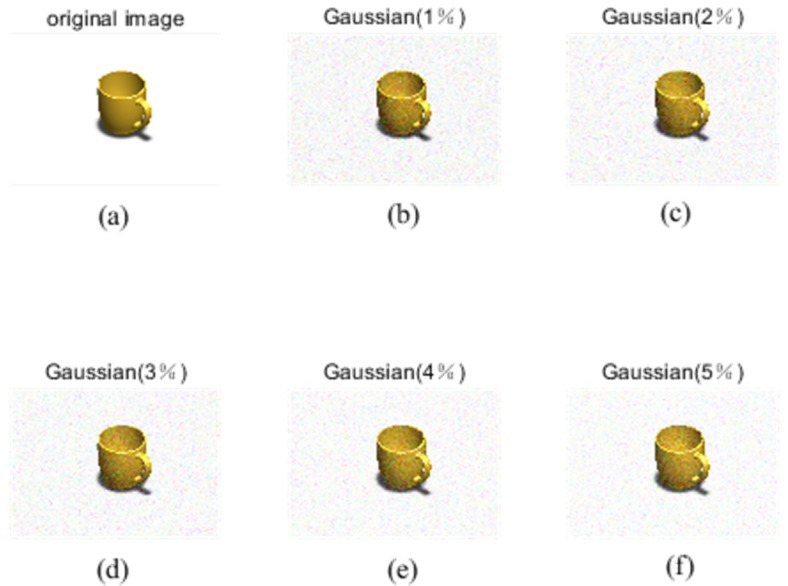
Gaussian noise plots for different densities of the PSB dataset.

**Figure 8 sensors-24-02352-f008:**
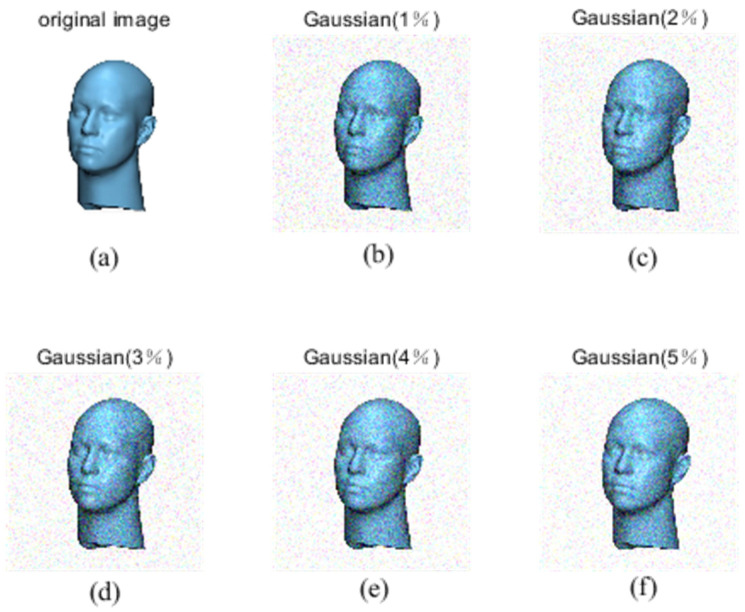
Gaussian noise plots for different densities of medical image dataset.

**Figure 9 sensors-24-02352-f009:**
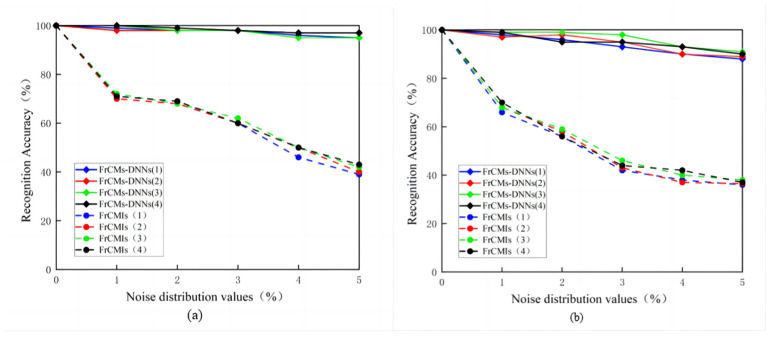
Comparison of the recognition accuracy of FrCMs-DNNs and FrCMIs with different score parameters: (**a**) the PSB dataset; (**b**) the medical images dataset.

**Figure 10 sensors-24-02352-f010:**
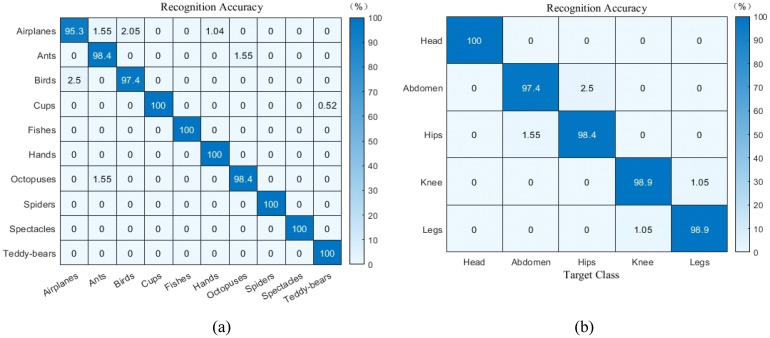
The confusion matrix of object recognition: (**a**) the PSB dataset; (**b**) the medical images dataset.

**Figure 11 sensors-24-02352-f011:**
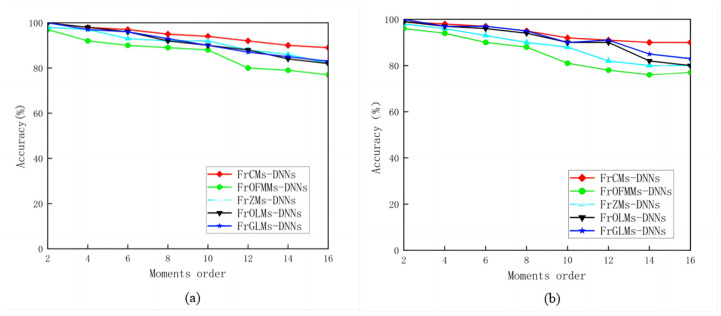
Comparison of the recognition accuracy: (**a**) the PSB dataset; (**b**) the medical images dataset.

**Figure 12 sensors-24-02352-f012:**
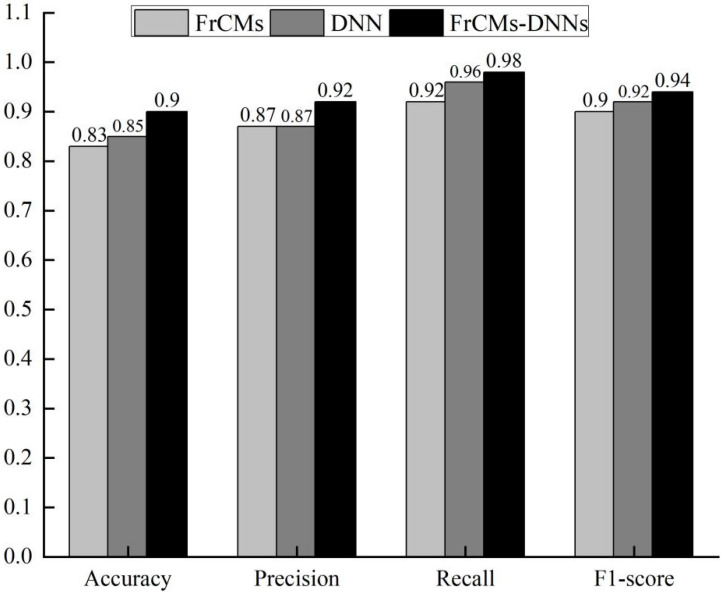
PSB dataset 3D object recognition results.

**Figure 13 sensors-24-02352-f013:**
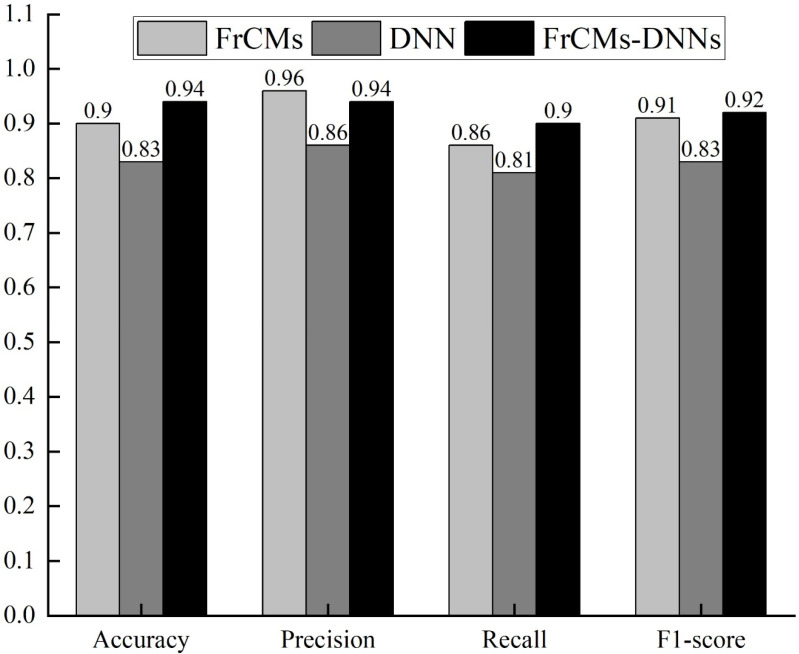
3D object recognition results from medical images dataset.

**Figure 14 sensors-24-02352-f014:**
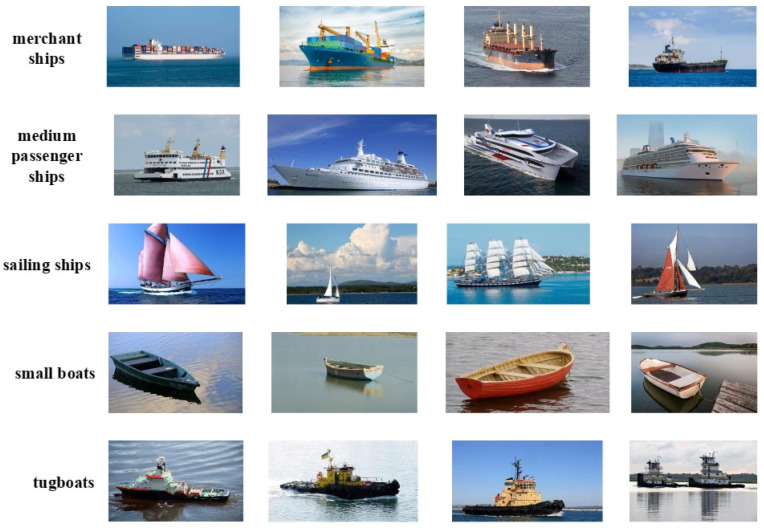
Category example images from the VAIS ship dataset.

**Figure 15 sensors-24-02352-f015:**
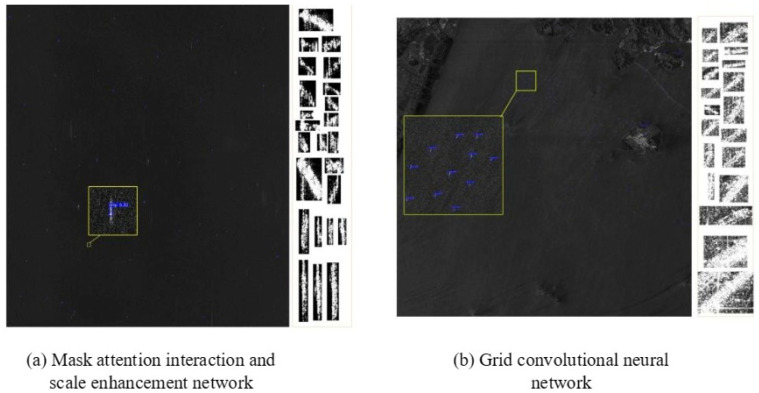
Chart of the high-speed SAR image ship detection results’ comparison.

**Figure 16 sensors-24-02352-f016:**
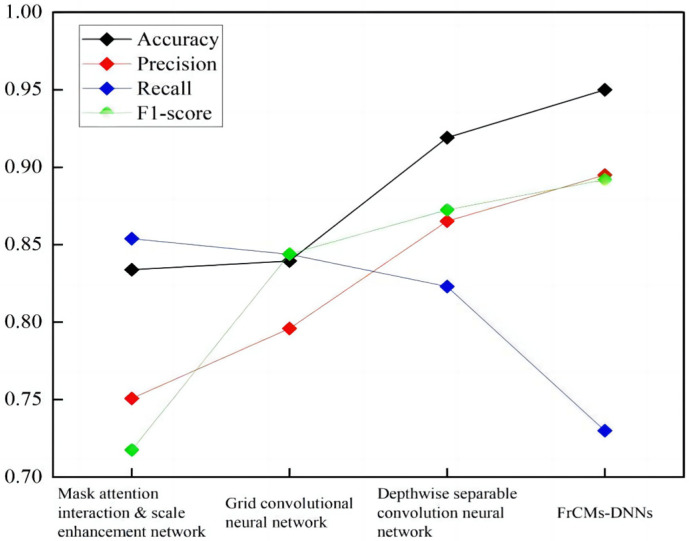
Comparison of accuracy, precision, recall, and F1-score of different network models.

**Figure 17 sensors-24-02352-f017:**
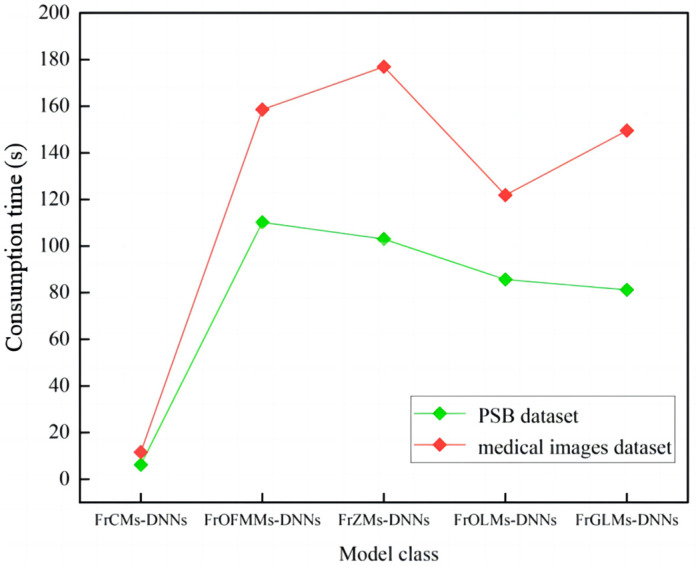
Comparison of consumption time of different classes of fractional-order moments–DNNs models under PSB dataset and medical images dataset.

**Table 1 sensors-24-02352-t001:** The detailed description of the FrCMs-DNNs’s structure.

Input Layers	Input Moment Vector	Activators N×N×N
1	Full Connection + BN + ELU + Dropout	100
2	Full Connection + BN + ReLU + Dropout	165
3	Full Connection + BN + ReLU + Dropout	245
4	Full Connection + BN + ReLU + Dropout	120
Output	Softmax	Quantity Subjects

**Table 2 sensors-24-02352-t002:** Three-dimensional recognition using DNNs and fractional-order Chebyshev moments.

Model	Main Aspects	Datasets	Evaluation Index	Advantages	Limitations
3D-CNN [31]	3D object recognition	ModelNet40	Recognition accuracy	High identification accuracy	The model is sensitive to local feature learning, occlusion, and attitude changes
3D-encoder-predictor CNNs and shape synthesis [32]	DNN for 3D model classification	ModelNet40	Classification accuracy	High classification accuracy; high training complexity	The model is sensitive to small datasets and noise
FrCMs [33]	FrCMs combined with DNNs for 3D recognition	ShapeNet	Classification accuracy and robustness	FrCMs provide better characterization, which reduces risk of overfitting	Selection and adjustment of FrCM parameters are more complex
FrCMs [34]	3D shape analysis	ModelNet	Segmentation accuracy and computational efficiency	The fractional order can accommodate a wide range of data distributions; it enhances the robustness	The computational cost of FrCMs is relatively high
FrCMs combined with 3D-CNN [35]	3D shape analysis	ShapeNet	Segmentation accuracy robustness	FrCMs enhance the understanding of shape structure; 3D-CNN extracts higher-level features	The selection and adjustment of FrCM parameters is complicated
DNN–FrCMs joint optimization [36]	Joint optimization of DNNs and FrCMs	3DShapeNet	Overall performance indicators	Combines the powerful modeling capabilities of DNNs with the feature extraction advantages of FrCMs	Requires significant computational resources for training
3D-CNN in association with FrCMs [37]	Application of fractional-order features to 3D scene understanding	Sun RGBD	Semantic segmentation accuracy	Comprehensive use of deep learning and fractional features; good adaptability to complex scenes	It takes a lot of computing resources to train

**Table 3 sensors-24-02352-t003:** Comparison of object recognition rates for the PSB dataset.

Moment Invariance	Noiseless	Gaussian Noise (1–5%)					Mean Value
		1%	2%	3%	4%	5%	
FrCMIs(1)	99.95	71.72	68.50	59.65	48.12	39.10	57.418
FrCMIs(2)	99.88	72.20	68.79	59.89	49.53	39.74	58.03
FrCMIs(3)	99.97	73.50	68.92	60.50	49.33	40.80	58.61
FrCMIs(4)	99.98	72.42	69.57	59.96	49.76	40.68	58.478
FrFMMIs	80.38	58.60	40.15	35.90	30.10	20.92	37.134
FrLMIs	98.30	66.30	58.95	50.25	36.32	30.87	48.538
FrGLMIs	97.87	65.79	57.90	53.48	35.92	29.75	48.568
FrZMIs	82.55	59.56	46.28	42.41	34.44	26.25	41.788
GMIs	75.60	33.23	23.35	18.85	16.70	14.57	21.34

**Table 4 sensors-24-02352-t004:** Comparison of object recognition rates for the medical images dataset.

Moment Invariance	Noiseless	Gaussian Noise (1–5%)					Mean Value
		1%	2%	3%	4%	5%	
FrCMIs(1)	99.35	66.15	54.76	42.71	36.57	32.76	46.59
FrCMIs(2)	99.92	67.36	56.45	45.45	35.70	33.84	47.76
FrCMIs(3)	99.92	67.40	57.25	46.89	40.05	37.16	49.75
FrCMIs(4)	99.37	69.57	55.01	44.52	41.10	36.16	49.272
FrFMMIs	79.90	46.85	34.59	30.85	27.02	21.35	32.132
FrLMIs	97.75	56.24	53.65	40.85	39.90	30.58	44.244
FrGLMIs	96.74	53.56	50.37	42.45	38.70	31.45	43.306
FrZMIs	81.90	45.02	34.15	30.91	29.10	22.12	32.26
GMIs	74.35	34.35	25.47	18.35	17.05	15.33	22.11

**Table 5 sensors-24-02352-t005:** Comparison of CPU computing efficiency of FrCMs-DNNs model.

Datasets	#.Param.	CPU FLOPS	Training	Inference	Top-1	Top-5
PSB	24.40 M	3.86 G	1024 FPS	1850 FPS	75.20	92.30
Medical Images	24.40 M	3.87 G	958 FPS	1680 FPS	77.52	93.35

**Table 6 sensors-24-02352-t006:** Comparison of GPU computing efficiency of FrCMs-DNNs model.

Datasets	#.Param.	CPU FLOPS	Training	Inference	Top-1	Top-5
PSB	24.40 M	7.34 G	2538 FPS	3905 FPS	78.95	94.55
Medical Images	24.40 M	7.35 G	2365 FPS	3685 FPS	78.52	93.89

**Table 7 sensors-24-02352-t007:** Accuracy, precision, recall, and F1-score for different categories of VAIS ship dataset.

Evaluation Index	Merchant Ships	Medium Passenger Ships	Sailing Ships	Small Boats	Tugboats
Accuracy	0.8539	0.8395	0.9191	0.8497	0.9500
Precision	0.8950	0.7508	0.8652	0.8400	0.5000
Recall	0.8539	0.5300	0.9191	0.8497	0.9500
F1-score	0.8725	0.7175	0.8920	0.8440	0.6550

**Table 8 sensors-24-02352-t008:** The time consumption of 3D object recognition for PSB dataset (unit: second).

Order(n,m,k)	FrCMs-DNNs	FrOFMMs-DNNs	FrZMs-DNNs	FrOLMs-DNNs	FrGLMs-DNNs
(0,0,0)	0.020	0.284	0.108	0.060	0.085
(2,2,2)	0.150	0.765	0.520	0.655	0.592
(4,4,4)	0.433	8.250	6.017	7.650	5.520
(6,6,6)	0.840	26.270	18.120	19.755	17.529
(8,8,8)	1.950	45.382	35.460	39.155	30.367
(10,10,10)	4.660	78.500	80.230	62.832	60.630
(12,12,12)	8.200	114.725	119.735	94.735	90.670
(14,14,14)	14.025	197.115	222.100	170.235	185.420
(16,16,16)	25.088	520.521	445.150	375.850	340.150
Mean value	6.152	110.201	103.049	85.659	81.218

**Table 9 sensors-24-02352-t009:** The time consumption of 3D object recognition for medical images dataset (unit: second).

Order(n,m,k)	FrCMs-DNNs	FrOFMMs-DNNs	FrZMs-DNNs	FrOLMs-DNNs	FrGLMs-DNNs
(0,0,0)	0.045	0.520	0.230	0.085	0.189
(2,2,2)	0.230	1.395	1.120	0.920	1.279
(4,4,4)	0.855	15.170	10.785	10.980	10.952
(6,6,6)	1.635	48.172	38.520	27.850	38.953
(8,8,8)	3.755	85.575	73.242	55.520	65.785
(10,10,10)	8.585	144.048	172.661	89.012	131.387
(12,12,12)	15.605	210.450	252.406	134.205	196.455
(14,14,14)	25.000	361.565	493.210	247.985	400.520
(16,16,16)	48.305	560.150	550.175	530.220	500.658
Mean value	11.557	158.561	176.928	121.864	149.575

## Data Availability

The raw data supporting the conclusions of this article will be made available by the authors on request.

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
