# Peer review of "Recognition of 3D Images by Fusing Fractional-Order Chebyshev Moments and Deep Neural Networks"

_sensors, 2024, doi:10.3390/s24072352_

Round 1

Reviewer 1 Report

Comments and Suggestions for Authors

In this article the author presents an image reconstruction and recognition method based on Chebyshev fractional moments and the DNN.

1. The title must be changed: we do not do the reconstruction with the DNN.

2. The model proposed by the author based on the fusion of FRCMs and the DNN is used for recognition and not for reconstruction.

3. The objectives of the author's work are not clear, the author must specify whether he will deal with reconstruction or classification?

4. The author did not present any problems in his work.

5. Paragraph of line 293-297 is duplicated paragraph of line 298-302

6. The presentation should be reviewed 2, 2.1, 2.1.1 and not (1)

7. Equations 1 to 21 have already been published in the literature, the authors of these equations must be cited.

8. The author must justify the choice of moments of Fractional Chebyshev invariant moments while there exist other moments such as: Legendre, Chebyshev, Laguerre, Hermite, and others

9. Most of the figures are blurry and unreadable, the author must redo all these figures

10. The author must cite the DNN reference

11. Table 1 line 1 the author must explain the term AB and the term ELU

12. The model proposed by the author has 3D moment matrices as input. To do the learning, the author must transform all the 3D image base into a 3D moment matrix base and annotate them. Is this task very heavy to carry out? I don't know how the author solved this problem.

13. After inputting the DNN model, the system performs convolutions on the moment matrix? The author should explain what type of convolution were used?. Then we must apply pooling to reduce the information. At the end we will end up with a vector of data which are very far from the initial values of the moments. The author must show that there is no loss of information on the vector of moments which are also characteristics of the images.

14. In my opinion, the author with this model seeks to extract the characteristics of an image from characteristics which are the moments. The author must explain the interest of applying a convolution on a feature matrix?. By applying pooling, will we not degrade the information contained in these characteristics?

15. The author confuses characteristics with reconstructions. Indeed, the title of Figure 4 is false. Figure 4 represents the reconstruction of images at different orders for different alpha parameters?

Comments on the Quality of English Language

Minor editing of English language is required

Author Response

Thank you for your very valuable suggestions. We attach great importance to your comments and have made feedbacks and modifications. In the feedback document, we provide relevant and detailed explanations of some issues. Please see the attachment.

Thanks again!

Best Regards!

Reviewer 2 Report

Comments and Suggestions for Authors

In this paper, the authors propose a deep neural network for 3D image recognition also exploiting the discriminating capabilities of the fractional order Chebyshev moments. Overall, the paper is well-written and results are encouraging. However, before publication, some modifications are needed.

For completeness, at the beginning of the Introduction some other applications of image moments (especially Chebyshev) can be quoted. For instance, radar target recognition [1]-[2], SAR target recognition [3], and sound classification [4].

[1] Pallotta L., et al. "Classification of micro-Doppler radar hand-gesture signatures by means of Chebyshev moments." IEEE MetroAeroSpace 2021.
[2] Machhour S., et al. "A Comparative Study of Orthogonal Moments for Micro-Doppler Classification." 2018 26th European Signal Processing Conference (EUSIPCO). IEEE, 2018.
[3] Bolourchi P., et al., "Target recognition in SAR images using radial Chebyshev moments". SIViP 11, 1033–1040 (2017). https://doi.org/10.1007/s11760-017-1054-2
[4] Neri M., et al., "Low-Complexity Environmental Sound Classification using Cadence Frequency Diagram and Chebychev Moments." IEEE ISPA 2023.

All figures in the paper appears blurred and so difficult to read. Please, use png format or another high quality format for figures.

Rather than showing the photos of ships in Figure 14 it would be more useful to see examples of SAR images for each class.

For SAR images, I suppose the authors are using their modulus as starting point for the algorithm. However SAR images are not 3D rather 2D. Therefore, I suggest removing this part. In fact, this goes behond the scope of this paper and results should be compared with a lot of method developed in the "SAR recognition literature".

Please, add the unit of measure for time consumption in Tables 8 and 9.

Comments on the Quality of English Language

I would recommend that the authors make a careful proofreading of the paper before next submission to improve English fluency

Author Response

Thank you for your valuable comments. We take your feedback very seriously and have responded and revised it on a case-by-case basis. Please see the attachment.

Thanks again!

Best Regards!

Round 2

Reviewer 1 Report

Comments and Suggestions for Authors

All comments and suggestions have been taken into account by the author in this revised version. Therefore I give a favorable opinion to publish the article